# The Mineral Oil Hydrocarbon Paradox in Olive Pomace Oils

**DOI:** 10.3390/foods12030434

**Published:** 2023-01-17

**Authors:** Raquel B. Gómez-Coca, María del Carmen Pérez-Camino, Wenceslao Moreda

**Affiliations:** Department of Characterization and Quality of Lipids, Instituto de la Grasa–CSIC, Ctra. Utrera km 1, Building 46, E-41013 Sevilla, Spain

**Keywords:** GCxGC-TOF/MS, hydrocarbons, mineral oil, MOSH–MOAH, olive pomace oil, online HPLC-GC-FID

## Abstract

The aim of this work was to understand the actual content of mineral oil hydrocarbons (MOH) in olive pomace oil in order to contribute to the monitoring requested by EFSA for the food groups making a relevant impact on human background exposure. Such information will complement both the information inferred from the limits established by the EU and the interpretation of the coming toxicological risk assessment. At the same time, the origin of such a group of compounds is discussed. From the raw material to the commercial product, olive pomace oils were sampled and analyzed at different points and/or conditions. Through the ultimate online HPLC-GC-FID system, we gathered information on the MOH concentrations and molecular mass profiles (C-fractions), and through GCxGC-TOF/MS, we identified the key structures that prove the innocuousness of the mineral oil aromatic hydrocarbon (MOAH) fraction. Our approaches provided chromatographic signals on the C10-C50 range, rendering 33–205 mg/kg mineral oil saturated hydrocarbon (MOSH) and 2–55 mg/kg MOAH in the commercial product. The results confirmed that the C25-C35 cut is the main fraction to which humans are exposed via olive pomace oil, showing concentrations highly dependent on the extraction process. Moreover, the identification of the main MOAH groups showed that in olive pomace oil, mainly 1- and 2-ring species were present, being virtually free of the carcinogenic 3–7 ring aromatics.

## 1. Introduction

According to the International Olive Council (IOC), olive pomace oil is the ‘oil obtained by treating olive pomace with solvent or other physical treatments, to the exclusion of oils obtained by re-esterification processes and of any mixture with oils of other kinds’. It may be labelled as crude olive pomace oil, refined olive pomace oil, or olive pomace oil composed of refined olive pomace and virgin olive oils [1].

In 2008, the Rapid Alert System for Food and Feed (RASFF) was alerted that several shiploads of sunflower oil adulterated with mineral oil had been exported from Ukraine to a number of Member States. As a result, the European Commission established a double control system according to which all Ukrainian shipments were to be tested in their own country prior to export as well as on the EU side when imported in order to assure that they did not contain more than 50 mg/kg mineral paraffins [2,3], indicating the contaminating nature of mineral oils, i.e., their identification as harmful substances actively introduced into the oil. Although these measurements were repealed in 2014 [4], the incident boosted the development of the methods of analysis focused on mineral oils and their products [5,6]

Nowadays, the ubiquity of mineral oil hydrocarbons (MOH) might raise certain questions, and here is where the paradox shows up: What if these substances are not in fact introduced into the oil—at least not in the active way one may think of when considering contaminants—but are naturally there? What if the anthropogenic character of MOH in olive pomace oils is not so straightforward because they are originally in the plant from which the oil would be extracted later on? Would they then be considered as actual contaminants or just as one of those groups of compounds whose presence is highly bound to the way the oil is extracted, as it in the case of, e.g., aliphatic alcohols, wax esters, sterols, or triterpenic dialcohols? In fact, the endogenous presence of the so-called unresolved complex mixtures in olive pomace oils was demonstrated in previous studies [7], although overlooking the aromatic fraction.

MOH (comprising MOSH and MOAH) come from crude mineral oil of petrogenic origin. It is formed by sedimentation of plant material at 5–6 km depth in the Earth’s crust, where at temperatures below 150–200 °C, it gives rise to the raw oil. Once the raw oil is extracted, it passes through a number of processes such as refining, hydrogenation, cracking, and distillation to produce derivatives such as diesel oil, jet fuel, or lubricants [8]. Regardless of the group, MOH consist of exceptionally complex mixtures with a high variability in C-atom number and structure. The composition of MOH depends on the oil of origin; the treatment applied after the crude extraction; and the possible addition of hydrocarbons from other sources such as coal, natural gas, or biomass [3].

Generally speaking, plants possess endogenous hydrocarbons, i.e., hydrocarbons of internal origin: Fruits have n-alkanes at percentages that vary from traces to 90% of the specific fraction [9]. Likewise, hydrocarbons are present in olive fruits, being the natural n-alkane series mainly concentrated in the pulp and those of mineral origin in the cuticle, the stone, and the leaves [7]. As a consequence, the presence of these kinds of compounds has also been described both in olive oil and in olive pomace oil at concentrations that vary as a function of the olive fruit variety and the extraction procedure [7]. MOH may also be found at different concentrations in almost all foods, both as a result of environmental depositions and from intentional uses in the food production chain [10].

The consequences of the human exposure to MOH are still an object of debate. They do not cause acute toxicity after intake, although the low biotransformation rates of some branched- and cyclo-alkanes makes the C16-C35 MOSH accumulate in different points of living organisms, initiating the formation of micro-granulomas whose consequences are still unclear [11]. MOAH, nonetheless, are well absorbed, not accumulated, and are easily disseminated to all organs, where they are widely metabolized depending their toxicological effects on the degree of both aromaticity and alkylation [12]. Therefore, it is important to identify the presence of potentially genotoxic species such as those with at least three aromatic rings [3,11,13].

Aware of the situation, the EU Commission requested Member States to collect data on the MOH content of different food groups in order to have enough information for future risk assessments. From the survey on which EFSA Scientific Opinion [3] was based, it was concluded that the food classes to be monitored must be selected on the basis of their contribution to the background exposure of the different sectors of population. Vegetable oils were included as one of the foods in which the MOH levels were the highest, with the background exposure to MOAH through such media being approximately 30–35% of the exposure to MOSH [3,10].

The lack of consensus regarding both the method of analysis and the data that must be reported, the great technical demands of the analytical approaches, the high uncertainty of the calculation methodology, etc., have led to numerous gaps in the pool of data available to EFSA in 2012, which led to the lack of a regulatory framework and therefore to limits for MOH in food products in the EU. Consequently, the food industry is walking blind in terms of many of the aspects related to these important contaminants, and so is the scientific community when it comes to advice regarding, e.g., mitigation strategies. The next EFSA draft, initially planned for the end of 2022 [14], and now for the beginning of 2023, may help to fulfil this gap, although it is mandatory to provide it with as much reliable data as possible.

Therefore, the significance of this work relies on its support to the olive pomace oil industry and to the science community intertwined with it, since it will contribute to EFSA’s data collection and EU monitoring by gathering information not only on the presence of MOSH and MOAH in commercial olive pomace oils but also at different stages of the production process starting with sampling open-air ponds at different points of time, and following with the influence of the extraction system and of the consecutive stages of the chemical refining process, all in all constituting an innovative approach to this situation. Finally, the absence of the alkylated 3–7 aromatic-ring hydrocarbons is also discussed.

## 2. Materials and Methods

### 2.1. Chemicals

Deionized water was of the highest purity possible. Ethanol absolute was purchased from VWR Chemicals (VWR Internationals, LLC, Llinars del Vallès, Barcelona, Spain). Sodium carbonate was from Honeywell Riedel-de Haën (Honeywell Specialty Chemicals, Seelze, Germany). 3-Chloroperbenzoic acid (mCPBA), dichloromethane (SupraSolv^®^, purity ≥ 99.8%, Merck KGaA, Darmstadt, Germany), n-hexane (SupraSolv^®^, purity ≥ 98%, Merck KGaA, Darmstadt, Germany), and sodium thiosulfate were obtained from Sigma-Aldrich (Merck KGaA, Darmstadt, Germany). MOSH–MOAH standards (reference 31070) and retention time standards (reference 31076) were from Restek (Restek SRL, Madrid, Spain).

### 2.2. Samples

The following samples were directly sent to our laboratory by the corresponding producers: 21 samples of crude olive pomace oil (COPO), 6 samples of neutralized olive pomace oil (OPO), 7 samples of winterized/washed OPO, 7 samples of bleached OPO, and 7 samples of refined OPO.

A total of 51 samples of commercial OPO as designated by the IOC (IOC, 2022), that is, oil composed of refined olive pomace oil (ROPO) and virgin olive oil (VOO), were purchased in supermarkets all over the country.

### 2.3. Sample Preparation and Epoxidation Procedure

Only glass or polyethylene terephthalate (PET) containers and glass or stainless-steel laboratory materials, inert for mineral oil, were utilized. They were all washed before use with the same n-hexane that was utilized during the analysis and dried at 300 °C overnight.

The epoxidation procedure was based on the methods described by Biedermann et al. [15] and by Nestola and Schmidt [16], although with modifications: oil (300 mg) was weighed into a 10 mL glass vial using the analytical balance (0.1 mg accuracy) and dissolved in 50 µL standard solution (1 mL Restek 31070 taken to a final volume of 10 mL with n-hexane) together with 600 µL n-hexane; 1 mL 10% mCPBA solution in ethanol absolute was then added. Everything was shaken at 3200 rpm for 15 min. Then, the epoxidation reaction was stopped by adding 1 mL aqueous 5% Na_2_CO_3_ + 5% Na_2_S_2_O_3_ solution. This preparation was again shaken for 2 more minutes, and then centrifuged at 3000 rpm for 5 min.

The upper phase (100 µL) was further analyzed (Section 2.4).

### 2.4. Instrumentation

MOSH and MOAH were extracted from the sample matrix using an organic solvent (n-hexane) after the addition of both internal and verification standards. Previous interferences (olefins) were removed following the epoxidation procedure described in Section 2.3. This preparative stage was performed manually if absolutely needed or with a Gerstel automatic system (Gerstel GmbH and Co. KG, Mülheim an der Ruhr, Germany) to minimize accidental contaminations.

The routine chromatographic analysis was carried out with an online coupled HPLC-GC-FID system where the MOSH and MOAH fractions were transferred automatically between the liquid and the gas chromatography columns, resulting in a higher sensitivity and a lower risk of contamination [17]. The system consisted of an Agilent 1260 Infinity II HPLC with UV-detector (230 nm) (Agilent Technologies, Santa Clara, CA, USA) through which the MOSH and MOAH fractions were separated. Each fraction was then transferred into a large volume (300–500 µL) to the combined double-channel Agilent 8890 GC. Here, solvent vapors were discharged through a solvent vapor exit placed between the uncoated pre-columns and the GC analytical columns. These specially designed evaporation techniques had initially been described by Grob [18]; in this system, volatile constituents are collected by solvent trapping, applying partially concurrent eluent evaporation and high boiling components spread over the complete length of the flooded zone to be then refocused by the retention gap technique [19].

HPLC assays were carried out by injecting 100 µL sample in a LiChrospher Si 60 silica gel column (5 μm, 250 mm × 2 mm ID; Merck KGaA, Darmstadt, Germany) using a n-hexane/dichloromethane gradient (30 min run). The initial flow was 0.30 mL/min, starting with 100% n-hexane for the elution of the MOSH fraction, and then with 35% dichloromethane for the elution of MOAH. The UV detector was utilized to monitor the HPLC separation. Through this column, complete separation between MOSH and MOAH can be achieved as long as the n-hexane used as an eluent is free of polar impurities [20]. After each run, the system back-flushes the column with dichloromethane (0.5 mL/min, 9 min) and then reconditions it with n-hexane (0.5 mL/min, 15 min).

GC analytical columns were provided with guard columns. These GC pre-columns were non-coated, deactivated Hydroguard MXT (10 m × 0.53 mm; Restek), whereas the GC separation columns were MXT-1 (100% dimethylpolysiloxane; 15 m × 0.25 mm ID × 0.10 μm film; Restek). Hydrogen was used as a carrier gas with an initial flow of 5.0 and 4.4 mL/min at the front (MOSH) and back (MOAH) channels, respectively. The oven temperature program started at 55 °C (10 min), and then it rose to 370 °C (20 °C/min) where it remained for 4.75 min. The temperature of both FID was 380 °C.

GCxGC analysis was carried out in a Pegasus BT 4D GCxGC-TOF/MS (LECO) with a cryogen-free thermal modulator (i.e., it does not need expensive cryogenic consumables). This kind of approach was carried out as a confirmation tool for MOSH and MOAH analysis following the recommendations of the Joint Research Centre (JRC) guidance [19]. Of the fractions from the HPLC collected from the transfer valve of an HPLC-GC system identical to that described above, 5 μL was injected into the GCxGC instrument using splitless mode. For such a system, a reverse set-up [17] was chosen in which the first-dimension column was a Rxi-17Sil MS (50% phenyl methyl polysiloxane, 12 m × 0.25 mm i.d. × 0.25 μm, Restek), connected to a Rxi-1 MS (100% dimethyl polysiloxane, 1.4 m × 0.15 mm i.d. × 0.15 μm) to perform the second separation. Uncoated, deactivated 0.53 mm i.d. (Axel Semrau, Sprockhövel, Germany) and 0.15 mm i.d. (Restek) guard columns were used in both dimensions, individually. The GC oven temperature program started at 60 °C (8 min), then rose at 5 °C/min until 350 °C (8 min); the transfer line temperature was 330 °C. The system was linked to the TOF-MS that worked with a constant flow of 1.5 mL/min helium as carrier gas. Modulation was performed every 5 s, applying variable hot and cold pulse durations on the basis of the wide range of volatilities in the sample. MS parameters: 50–600 *m*/*z*; 30 kHz spectra generation frequency; interface and ion source temperatures of 250 °C and 340 °C, respectively; electron ionization (EI) at 70 eV. Data were acquired and elaborated with the LECO ChromaTOF software [21].

### 2.5. Quality Assurance and Quantitative Analysis

Quality assurance and quantitative analysis were carried out using the Restek 31070 standard mixture. This mixture consists of

n-Undecane (C11), verification standard, added at the same concentration as cyclohexylcyclohexane (CyCy) to keep track of possible losses of both volatile compounds and CyCy.

CyCy, quantification standard and not present in substantial concentrations in mineral oils. In a non-polar GC column, it is eluted in the MOSH fraction between C11 and n-tridecane (C13).

C13, second verification standard, added at half concentration of CyCy, forming with it a straightforwardly recognized pair that allows for the checking for possible co-elution with CyCy.

Cholestane (Cho), which signals the end of the MOSH fraction.

The standards for MOAH (pentyl benzene,5B; 1- and 2-methylnaphthalene, 1- and 2-MN; and perylene, Per) worked analogously. Additionally, 1,3,5-tri-tert-butyl benzene (TBB) marks the beginning of the elution of the fraction.

The Agilent OpenLAB for GC program was used for data acquisition. Chromatogram integration and final calculation were carried out with the Gerstel Enterprise Edition program. The signal area in the FID chromatogram corresponding to MOSH–MOAH was calculated by the integration of the chromatogram covering the range of ≥ n-C10 to ≤ n-C50, subtracting the corresponding blank baseline run on every sample batch. As described previously [7], mineral oil components are determined by the area of the hump confined by the baseline and the riding peaks attributed to the naturally occurring n-alkanes (odd C-atom number species, mainly from n-C21 to n-C35) and hydrocarbons of terpenic origin. Then, the calculation of the MOSH and MOAH mass fractions is accomplished by using the equation described at the JRC guidance [19], defining the C-fractions in the chromatograms (‘cuts’) by the position of the chromatographic peaks of the corresponding n-alkanes (Restek 31076), taking into account that each of them starts at the retention time of the peak end of the starting n-alkane of the range and ends at the retention time of the peak end of the closing hydrocarbon, with the exception of the first sub-fraction, which begins at the retention time of n-C10.

## 3. Results and Discussion

### 3.1. On-Line HPLC-GC-FID

MOH analysis focuses on the C10–C50 range. There, MOSH and MOAH are independently determined as two separated groups. Within each of them, sub-classes are distinguished on the basis of molecular mass ranges and, in the case of MOSH, on structures. Data on MOH composition come mainly from GC, through which, even at its highest separation efficiency, MOSH render a pattern consisting of a hump from a mixture of components impossible to separate, with well-defined n-alkane peaks on top. The GC pattern of MOAH is similar, although with hardly any riding peak, depending on the sample nature and on efficiency of the sample pretreatment (Figure 1).

As pointed out above, before reaching the GC analysis, one or more cleaning or preparative steps are needed. Generally speaking, ‘difficult’ (i.e., high fat content) samples may require the saponification of the triacylglycerides (TAG) and removal of the resulting soaps [22,23,24,25]. The drawbacks of this approach are clear: it is tiresome, time consuming, and with increased chances of accidental contamination.

A second alternative is to rely only on LC, which may separate the MOH fraction from the surrounding lipids. It is therefore important to use an HPLC column with enough capacity to retain TAG. Furthermore, through HPLC, the separation of MOH into paraffins and naphthenes (MOSH), as well as aromatic compounds (MOAH), is carried out, but since there is no selective detector, this step is merely used for a pre-fractionation prior to the GC analysis. In HPLC-GC-FID systems such as that used here, such (HPLC) separation (for which there are not chromatographic traces) results in two different GC-FID chromatograms (Figure 1), since the equipment has two GC columns working in parallel.

Before passing through the LC column, it is advisable to eliminate potential interferences that may hamper the quantification of the GC hump. The type of interference depends on the group: On the one hand, the analysis of the MOSH may entail the removal of hydrocarbons of plant origin such as the odd C-atom number n-alkanes that may overload the GC column. This can be achieved with activated aluminum oxide, which retains long-chain alkanes (>C22), whereas iso-alkanes and naphthenes pass through [26,27]. On the other hand, the analysis of MOAH requires the removal of squalene, its isomerization products, sterenes, and derivatives of carotenes formed during oil refining. This can be achieved by selective epoxidation of the olefins, rendering them more polar and with a higher retention than MOAH by means of the formation of an oxirane ring [15,16,20,28]. MOAH chromatograms on Figure 1 show both high (right hand side) and low (left hand side) epoxidation efficiency, demonstrating that such pretreatment must be carefully adapted to each sample case.

Once the two fractions are ‘clean’ and separated, the next step is their GC-FID-based analysis. GC is the analytical technique of choice because it isolates the hydrocarbons naturally occurring in foods from those of mineral origin incorporated either during plant growth or through external pollutants. Within this last group, it is also possible to characterize compounds by molecular mass range, although without accomplishing resolution into individual constituents [20,29]. Since separation of individual compounds is not possible, short columns are the first choice in order to obtain the advantage of faster runs. Commonly, non-polar stationary phases are used, making analogous series of analytes separate according to their boiling point. Moreover, the molecular mass distribution of the different MOSH sub-fractions can be inferred from the identification of the matching n-alkane, knowing that branched species of the same mass elute at a retention time corresponding to an n-alkane of up to roughly two C-atoms less, and cycloalkanes do so at a retention time equivalent to that of an n-alkane of two C-atoms more. For MOAH, the relationship is not so straightforward.

The system we have depicted is nowadays considered as the method of choice for the quantification of mineral oil in routine analysis [17]. In our case, each run, including the preparative stage, takes around one hour, although samples are processed in a way that the cleaning of a certain sample overlaps with the chromatographic separation of the preceding one instead of being completely sequential, saving quite some time in a complete analysis.

The results of this work, next discussed, are reported in mg/kg, with two significant figures and rounded according to the rules given by ISO 80000 [30]. When analyzing these data, one must keep in mind that even though they were obtained through a method that complies with the performance parameters described by the JRC guide [19], they are affected by a high uncertainty whose main influence comes from the difficulty of the integration, especially from manually drawing the curve under the riding peaks and/or outlining the baseline.

#### 3.1.1. Effect of the Time in Open-Air Ponds

Table 1 shows the MOSH–MOAH content of ten different COPO samples numbered from 1 to 10 (column 1).

COPO comes from olive pomace (alperujo), a wet semi-solid by-product obtained in two-phase olive mills during olive oil extraction. Nowadays, those mills are improving their centrifugation systems to increase the quantity of olive oil extracted, leaving pomace with very low oil content, making its extraction not very attractive from an economic point of view. Alperujo is normally stored for a variable period of time (from several months to the end of the campaign) in large open-air ponds until it is processed in extraction plants. In the cases at hand, COPO labelled from 1 to 10 was obtained from pomace sampled from ponds sheltered from the rain, at medium depth, at one, two, or three different moments during the campaign (depending on the producer samples that were sent to the laboratory in August 2018, March 2019, and/or July 2019), with this being signaled with the letters ‘i’, ‘m’, or ‘f’, depending on whether the corresponding aliquot was taken at the beginning, in the middle, or at the end of the campaign, respectively. As we can observe, for a given COPO, there was no difference in the total MOSH–MOAH content, including in the content of the corresponding C-fractions between consecutive sampling times, considering the 20% method’s uncertainty [19]. Therefore, atmospheric contamination during storage in open-air ponds is not the main source of the hydrocarbons present in COPO, as has already been demonstrated [7]. The same phenomenon was described by Moret [31] who measured MOH concentrations during pomace storage between 17 and 300 mg/kg, discarding contamination during both storage and mechanized handling. Moreover, the possibility of the formation of MOH, e.g., by fermentation, was rejected, supported by the presence of hopanes (Figure 2), accepted markers of mineral oils [32]. In fact, steranes and/or hopanes were detected in all commercial samples under analysis (Appendix A).

#### 3.1.2. Influence of the Extraction System

Table 1, column 2, specifies the extraction process. Clearly, all those producers that had applied centrifugation (physical extraction) obtained COPO with much lower MOSH–MOAH content that those in which solvent had been utilized: 65–118 mg/kg MOSH and 1.8–28 mg/kg MOAH, and 209–520 mg/kg MOSH and 54–115 mg/kg MOAH, respectively, all of them mainly concentrated on the C25–C35 range, followed by the C35–C40 fraction in the case of MOSH, and by the C16–C25 cut for MOAH. The results agree with that of previous studies in which the saturated hydrocarbon concentration ranged from 16 to 140 mg/kg for physical extraction, but, however, went from 100 to 300 mg/kg with the use of solvent [7,31,32]. Globally considered, these results indicate that the use of solvent during the extraction process has the effect of concentrating saturated hydrocarbons between two and six times, whereas in the case of the aromatic species, the concentration may be from 4- to 30-fold. 

To rule out the possibility of MOSH–MOAH being transferred from the hexane to the pomace during the extraction, we analyzed three different hexane brands. The results showed MOSH and MOAH concentrations between 14 and 26 mg/kg, and between 5.5 and 12 mg/kg, respectively, for newly taken hexane aliquots (the molecular mass distributions were similar to those observed in the oil samples as in, e.g., Table 1). However, when repeating the measurements after using them in an extraction process, concentrations turned out to be 26 mg/kg MOSH and 7.4 mg/kg MOAH, signaling the neutral effect of the solvent in the global process. This might be explained by the existence of an equilibrium according to which small amounts of the MOH content of the solvent are transferred to the paste and vice versa.

The low MOSH–MOAH concentration in COPO when it has been extracted through centrifugation in comparison with hexane extraction is due to the fact that an organic solvent at high temperature is a more efficient solution media for MOH than the matrix itself (TAG). Actually, under the standard time and temperature conditions applied to the mass of pomace during physical extraction, MOSH–MOAH were dissolved in a relatively small amount of oil, with a proportionally high quantity remaining in the exhausted solid. This pomace may be processed a second time, depending on the producer, rendering even smaller volumes of COPO with low amounts of hydrocarbons that, however, becomes concentrated in the remaining solid and therefore in the successive extracted oil fractions. Once the physical extraction process is over, the solid—once dried—can also be passed through hexane in order to obtain whatever oil might be left, and then both fractions are mixed. The result is again a COPO where the MOSH–MOAH concentration is noticeably higher than in those cases where only centrifugation has been applied. This outcome is due to both the higher efficiency of hexane as a MOH dissolving agent and the concentration effect produced after each extraction. The result of using physical and solvent extractions consecutively on the same pomace is clear when comparing the results for samples 6m–6f with those of the other samples, confirming our previous statement on the concentrating effect when using solvent for COPO extraction. In fact, when comparing the average values for MOSH and MOAH concentrations in oils obtained through centrifugation (78 and 12.3 mg/kg, respectively) with those of samples 6m–6f (343 and 72.5 mg/kg, respectively), one can observe that MOSH and MOAH were concentrated by 4.4 and 5.9 times, respectively.

As explained above, alperujo retains water, making it require a special treatment to obtain a product feasible for oil extraction through the use of solvent. After separating the main part of the stones pitted, alperujo is centrifuged in the decanter, which allows oil extraction also from low-fat material, rendering physically extracted COPO. The remaining solid still contains both water (60–65%) and oil. In order to extract as much fat as possible, the solid can be solvent extracted, for which it has to be previously dried to no more than 10% humidity [33]. In fact, the higher the water content in the pomace, the more complicated the oil extraction, and this is why the application of a thermal pre-treatment facilitates the process. The drying step is performed in rotary heat dryers known as tromels where alperujo is subjected to gases at high temperature (400–800 °C). The fact that these gases may be generated by burning exhausted olive pomace (orujillo) or olive stones, or by co-generation, made us wonder whether such a step had an influence on the MOH content of the samples, since at least most of the MOSH fraction in the olive fruit concentrates on the cuticle and the stones themselves [7]. Actually, it is known that sunflower seeds were contaminated during drying with an indirect heat source, although how occurs is still unclear [10,34,35]. Therefore, we decided to compare the MOSH–MOAH content of those samples for which such information had been provided from the extraction plant. In this way, we knew that samples 1 and 8, obtained by centrifugation, had previously been dried in a traditional oven using orujillo as biomass fuel, or by the use of a co-generation system using natural gas, respectively. No differences were observed between both systems on the total MOSH–MOAH content on neither of the individual C-fractions (data not shown). Regarding COPO obtained via solvent, different fractions of sample 7 (i.e., 7i, 7m, and 7f) dried with orujillo were compared with samples 4, 5, and 6, dried by co-generation, supporting the aforementioned observation. For these samples, the influence of the solvent extraction method itself could be observed, since sample 7 had been subjected to a continuous system where the pomace was placed on a conveyor over which hexane was sprayed, whereas samples 4–6 were extracted in a discontinuous process, i.e., both pomace and solvent were mixed together in large cylinders and then separated. Again, comparing both sets of samples, no differences on the analyte composition and content could be inferred.

After analyzing COPO, a mandatory question arose: Where does the detected MOH come from? On the one hand, the petrogenic nature of these compounds was demonstrated for all our samples, at least in part, through the determination of hopanes, diagenesis products of hopanoids formed during geological times at the temperatures reached during deep sediment burial [32]. On the other hand, its origin may be endogenous and/or attributable to external agents. The responsibility of atmospheric particles from the exhausts of vehicles, motor oils from the bulldozers utilized to move the pomace around the storage area, or lubricating and hydraulic oils applied to the extraction machinery was ruled out earlier [7]. That, the concentration range detected, the symmetry between the MOSH and MOAH chromatographic curves (Figure 1), and the molecular mass distributions make us think that the presence of MOH in the samples under study is due to soil depositions incorporated during plan growth and not to any external contamination source, therefore being unavoidable.

#### 3.1.3. Effect of Chemical Refining

Regardless of the method of extraction, COPO must be refined in order to be fit for human consumption. Moreover, the refining process can be chemical or physical, depending on the inclusion of a neutralization step with caustic soda at the beginning of the procedure to neutralize the free fatty acid present in the media. Generally speaking, refining allows for the elimination of peroxides, degradation products, unpleasant volatiles, and other minor compounds from the matrix, rendering a neutral, safer, and more stable product. Table 2 shows the effect of the different refining steps on seven samples. Except for sample 9f, for which data on the neutralized fractions were not sent by the producer, we determined the MOSH–MOAH concentration for the different C-fractions for the original (raw) oil and the oil fractions after neutralization, washing and/or winterization, bleaching, and deodorization (marked as ‘refined’). From these results, we can conclude that, regardless the possible differences among treatments from one producer to another, the refining process does not have any significant effect on the total content of MOH and that its concentration in ROPO will be that of the raw material. The C10–C16 fraction was the only one in which a certain loss was observed due to its high volatility. Since the contribution of this first segment to the total hydrocarbon content was minimal, its disappearance passed unnoticed.

Additionally, olive pomace oil producers can prepare batches of COPO from different origins. We checked the effect that this kind of practice may have on the MOSH–MOAH content by mixing two independent oil samples. Surprisingly, what we found was far from the expected diluting effect and close to an additive result, since from the original 337 and 279 mg/kg MOSH, and 81 and 68 mg/kg MOAH, respectively, our analysis revealed 556 mg/kg and 129 mg/kg MOSH and MOAH in the final blend.

#### 3.1.4. Commercial Olive Pomace Oil

Commercial olive pomace oil is a mixture of ROPO and VOO. Such oil is labelled as ‘OPO composed of refined olive pomace and virgin olive oils’ [1]. Knowing the role that vegetable oils have on consumer background exposure to MOH [3], it was also interesting to actually know the possible consumer intake to those compounds once they took the OPO normally found on the supermarket. Table 3 shows the results obtained after a survey carried out all over the country. Globally speaking, we can say that the content of MOH in olive pomace oil goes from 35 to 369 mg/kg, a range similar to that in grapeseed oils, from 43 to 247 mg/kg, coming predominantly from the rests of peels in the seeds [36]. From that, 33 to 205 mg/kg corresponded to total MOSH (i.e., C10–C50) and 2–55 mg/kg to total MOAH. Regarding the MOSH profile, only compounds above twenty carbon atoms were detected and only in thirty-six out of fifty-one samples, with concentrations between 1.1 and 12 mg/kg for the C20–C25 fraction. The C25–C35 was the prevailing C-cut, reaching concentrations between 15 and 139 mg/kg, followed by the C35–C40 fraction whose components had concentrations of 9.4–57 mg/kg; this is in agreement with values obtained from the literature where commercial olive pomace oil showed a MOSH concentration of 121–250 mg/kg [17,31], in the same order of magnitude than those in other edible oils such as the blend of olive oil with grapeseed oil, in which 259 mg/kg MOSH were measured [37], probably due to the pesticide formulation used in grape seeds [10]. Concerning MOAH, none of them had measurable C10–C16 fraction, and only sixteen had C16–C25 concentrations above the method’s LOQ (i.e., 1 mg/kg), being around 5.7 mg/kg in the upper limit. Again, C25–C35 was the main fraction with values between 2 and 27 mg/kg. The second most important C-cut was also C35–C40, only measurable in twenty-four samples at concentrations between 1 and 22 mg/kg. The total MOAH content (i.e., C10–C50) in the samples under study lay within the 2–32 mg/kg range. 

The tendency of the prevailing C25–C35 fraction followed by the C35–C40 cut seems to be a trend in vegetable oils. A recent publication [38] showed how MOSH concentrations in palm oil and in coconut oil fulfil such behavior (20–29 mg/kg and 35–42 mg/kg, respectively). However, this seems to be somehow different in matrices from animal origin. For instance, MOSH concentration of 9–20 mg/kg was measured in fish feed, with this being centered in the C25–C35 zone of the chromatograms, followed by the C20–C25 cut, meaning a different molecular distribution for this group of compounds. Regarding MOAH, its concentration in fish feed was 1–5 mg/kg, and it came from compounds equally distributed among all C-cuts, meaning values below the method’s LOQ were added up. Again, the situation was different in palm oil and in coconut oil, where the C35–C50 fraction was the predominant one with values of 3–6 mg/kg and 9–11 mg/kg, respectively [38].

### 3.2. GCxGC-MS/TOF

FID is the most suitable system to quantify MOH, mainly because it does not suffer from the calibration problems encountered with other methods (e.g., MS). In addition, it is the only detector available for the quantitative determination of complex mixtures of hydrocarbons for which there are no standards since it has an almost identical response for all of them, potentially also being seen as a lack of selectivity. That and its low sensitivity in the face of the broad patterns of unresolved/unidentified peaks formed by mineral oil products are not the best of situations [5] but it is more than enough for routine analysis.

If higher resolution is required, comprehensive two-dimensional GC may be the solution. In this way, a second, independent run along an extra GC column supplies an additional separation degree. In the reverse setup, the separation on the first dimension is performed in a low-polarity column, whereas the second dimension works with a non-polar one. The MOSH fraction of one of the commercial samples analyzed is shown in Figure 2. There, the intense cloud encircled in black corresponds to the unresolved, highly isomerized hydrocarbons of mineral origin (the non-encircled cloud below shows some kind of column overloading in the second dimension whose origin must be further investigated; both the lack of alox treatment and/or the wear of the chromatographic column may be possible causes). The predominance of the C25–C35 fraction is signaled in red. The places in which some distinctive compounds should be are marked with dotted lines; among them, the standards C13 and CyCy are clearly visible. Below the location of the (assumed) n-alkanes, a group of small clusters, probably corresponding to little cyclic-hydrocarbons (naphthenes), appear. As pointed out earlier, steranes (including the standard Cho) and hopanes are visible below the previous ones since in this configuration, the more rings a hydrocarbon has, the lower the position in the plot [17].

Figure 3B shows the corresponding MOAH fraction, and Figure 3A shows a mixture of mineral oil from different origins covering a broad mass range, used for comparisons, forming parallel bands of a given number of aromatic rings (the more rings, the lower the position in the plot as it would correspond to a rapid elution on the second, non-polar dimension) that extend through the chromatogram. From Figure 3B, one can deduce the absence of three or more aromatic-ring species, since the outcome was similar for all our samples (Appendix A), i.e., in all of them, there was a lack of high-intensity chromatographic signals below the pink dotted line. Instead, a high-intensity, unresolved (and therefore very alkylated) cloud appeared mainly in the zones of the mono- and diaromatics.

### 3.3. Remarks on MOAH Genotoxicity and Risk Assessment

Throughout this work, evidence of the lack of genotoxicity of the MOAH fraction of olive pomace oil are presented in two forms: First of all, the hump shape of the HPLC-GC-FID chromatographic signal demonstrated a lack of separation of this group in individual compounds, which occurred due to their high degree of alkylation—the higher the alkylation degree, the lower the genotoxic potential; potentially carcinogenic compounds are either ‘naked’ or poorly alkylated [39]. Second of all, the absence of structures with more than two aromatic rings was confirmed via GCxGC-TOF/MS.

Additionally, the endogenous nature of the MOAH fraction in the olive oil samples under study was hypothesized, supported by, on the one hand, the symmetry between the respective MOSH–MOAH humps (obtained through HPLC-GC-FID), meaning that if the first one is of endogenous origin, as was the case [7], the other is endogenous too. On the other hand, only MOAH-free, food-grade lubricants are used in the European Union food industry, but nonetheless, if technical-grade lubricants are utilized, they must comply with the IP346 test, which guarantees that the three or more aromatic-ring hydrocarbons in the MOAH fraction are not above 3%.

Polycyclic aromatic compounds (PAC) are a group of substances associated with carcinogenicity. However, not all PAC behaves the same, and only those species containing N or S heterocycles, or 3–7 aromatic rings (e.g., polycyclic aromatic hydrocarbons, PAH), are carcinogenic, whereas the highly alkylated 1–2 aromatic-ring compounds typical of refined oils (i.e., MOAH in olive pomace oil) are not bioactive and therefore not genotoxic. In order to distinguish both groups of compounds, the DMSO-based extraction known as IP346 has been applied for more than 30 years, being the only method accepted in Europe (and also in Australia, China, and the USA) to assess the carcinogenicity potential of lubricating base oils [39,40]. Moreover, edible oil safety is secured during refining thanks to the treatment with, e.g., active carbon, which eliminates PAH [41,42].

These facts call for an adequate risk assessment, completely different for non-genotoxic substances than for genotoxic compounds. Therefore, we caution against the agreement expressed by the Member States to withdraw oils from the market when the sum of the concentration of MOAH reaches 2 mg/kg [43], first of all because 2 mg/kg is the limit of quantification (LOQ-max) set by the JRC for fats and oils [19], and any measure made there will suffer from high uncertainty (20% intermediate precision), and second of all, because as demonstrated, not all compounds within an aromatic fraction have the same genotoxic potential or are genotoxic at all [12].

## 4. Conclusions

Low-quality oils such as pomace oils usually contain more waxes, sterols, and aliphatic alcohols than virgin olive oils since all of them are mainly located in the fruit skin. This is also the case of MOH, which rest in the semi-exhausted solid residue to be extracted together with the small amount of residual oil.

The level of MOH concentration in pomace oils does not evolve over the time in the ponds and directly depends on the extraction process, being noticeably higher if solvent extraction has been applied than with only a physical process such as centrifugation. The intermediate drying step, the way in which hexane is applied during the extraction, and the possible presence of MOH in the solvent itself do not show any influence.

The different steps in the chemical refining process do not alter the original amount of MOH in the oil, except for the case of the C10–C16 fraction, lost due to its high volatility. Such loss passes unnoticed due to its very low value and the high uncertainty of the method in the low concentration range.

MOH concentration (MOSH and MOAH) in commercial olive pomace oils, which is around 250 mg/kg, depends on the quantity of MOSH–MOAH in the raw material, which includes both the refining olive pomace oil and the virgin olive oil, which must be added prior to going to the market. All in all, there is no plausible explanation for the presence of MOH in olive pomace oils in general: the mechanized handling, use of white mineral oil to clean and maintain the oil mill machinery, talc used as aid during the extraction, etc., might explain part of the phenomenon in the cases of concentrations above the boundaries shown in this study, but not all. The question on the possibility of the incorporation of this group of substances (including MOAH) from the soil and the atmosphere to the plant during its development is under study and would complement the conclusions reached so far on their endogenous nature [7].

Regarding the MOAH structure, none of the 3–7 aromatic-ring compounds were detected through bidimensional gas chromatography. Moreover, only highly alkylated species came out in this fraction. Therefore, for the time being, one may say that there is no genotoxicity risk associated to olive pomace oil from this point of view.

Finally, it should be added that olive pomace oils are nothing out of the norm regarding the subject treated here. In fact, high MOH levels have been detected in numerous edible oils such as pumpkin kernel, 250 mg/kg; linseed, 450 mg/kg; safflower, 500 mg/kg; soybean, 1000 mg/kg; and nut, sesame, and other cold pressed oils in which concentrations above 1000 mg/kg have been measured [9]. Whether this is the usual trend or this is something exceptional—occurred in light of fraud—remains to be discussed since we do not have first-hand data to assure either factor.

## Figures and Tables

**Figure 1 foods-12-00434-f001:**
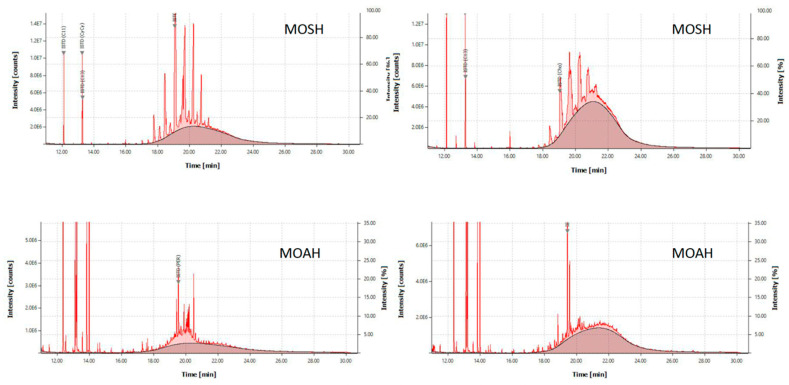
One-dimension HPLC-GC-FID traces of MOSH and MOAH in two (right and left) olive pomace oil samples. Hydrocarbons of plant origin and olefins can be seen on top of the MOSH and MOAH fractions, respectively. The darker areas correspond to the hydrocarbons of mineral origin.

**Figure 2 foods-12-00434-f002:**
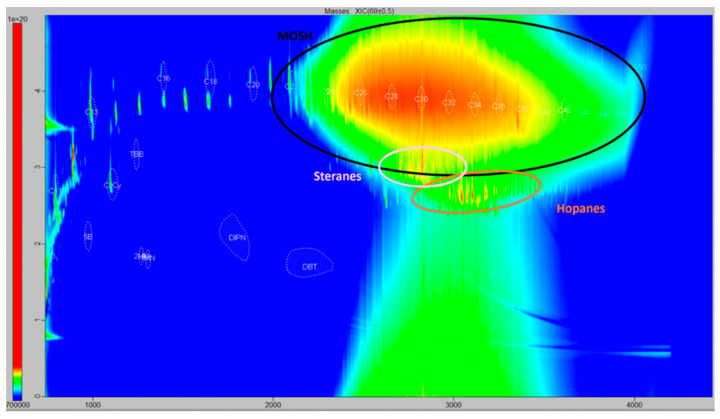
Extracted ion chromatogram of the MOSH fraction of an olive pomace oil sample, obtained by HPLC-GCxGC-TOF/MS, corresponding to *m*/*z* 69.

**Figure 3 foods-12-00434-f003:**
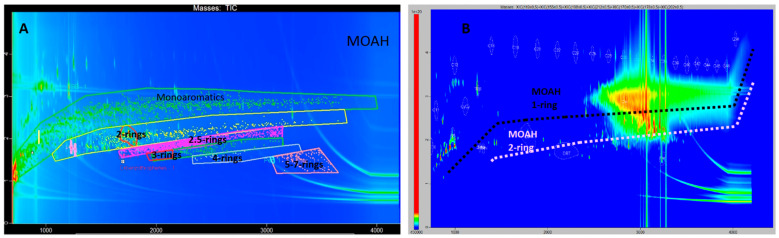
(**A**) Total ion current (TIC) two-dimension chromatogram for the MOAH fraction of a mixture of mineral oils from different origins, obtained by LC-GCxGC-TOF-MS, used for comparisons. (**B**) Extracted ion chromatogram of the MOAH fraction of an olive pomace oil sample corresponding to *m*/*z* 119, 155, 198, 170, 178, and 202.

**Table 1 foods-12-00434-t001:** Mineral oil saturated hydrocarbon (MOSH) and mineral oil aromatic hydrocarbon (MOAH) content in crude olive pomace oils sampled at the beginning (i), in the middle (m), and/or at the end (f) of the campaign and obtained through physical (centrifugation) and/or solvent extraction means. Results, in mg/kg and with two significant figures, are given for the C-fractions and for the sum of them.

Sample	Extraction	MOSH, mg/kg						MOAH, mg/kg				
#	Process	C10–C16	C16–C25	C25–C35	C35–C40	C40–C50	Total	C10–C16	C16–C25	C25–C35	C35–C40	Total
**1i**	Physical	1.1	4.7	47	12	4.4	**69**	0	2.0	4.8	0	**6.8**
**1m**	Physical	0	11	49	19	3.0	**81**	1.3	5.1	10	4.7	**22**
**1f**	Physical	1.1	6.0	46	9.2	3.0	**65**	0	0	1.8	0	**1.8**
**2i**	Solvent extr.	2.8	43	130	38	5.3	**219**	1.8	32	28	6.6	**68**
**2m**	Solvent extr.	1.3	52	151	29	8.5	**241**	2.4	21	29	2.3	**54**
**3i**	Physical	1.2	4.8	54	10	3.3	**73**	0	1.9	4.7	0	**6.6**
**3f**	Physical	1.1	4.6	38	10	4.3	**58**	1.1	3.7	2.0	0	**6.8**
**4i**	Solvent extr.	1.2	28	177	19	7.7	**234**	3.1	21	29	3.7	**57**
**4m**	Solvent extr.	2.7	41	172	37	5.4	**258**	3.2	32	35	7.1	**77**
**5i**	Solvent extr.	0	41	318	119	17	**495**	0	4.5	61	34	**100**
**5m**	Solvent extr.	4.2	88	359	60	8.8	**520**	4.9	54	44	11	**115**
**6m**	Physical+ Solvent extr.	3.8	78	218	56	7.0	**363**	4.6	39	38	7.6	**89**
**6f**	Physical+ Solvent extr.	1.8	72	190	59	0	**323**	2.9	30	21	1.9	**56**
**7i**	Solvent extr.	2.0	47	179	20	7.0	**255**	3.0	21	34	2.3	**61**
**7m**	Solvent extr.	1.3	40	130	38	5.2	**215**	1.8	34	30	7.1	**73**
**7f**	Solvent extr.	6.4	50	120	30	4.0	**209**	2.4	28	31	3.4	**64**
**8m**	Physical	1.2	21	48	12	2.2	**84**	0	1.5	10	2.2	**14**
**9f**	Solvent extr.	1.6	13	151	47	7.0	**220**	0	27	29	7.8	**63**
**10f**	Physical	1.5	18	76	16	6.5	**118**	0	10	14	4.1	**28**

**Table 2 foods-12-00434-t002:** Mineral oil saturated hydrocarbon (MOSH) and mineral oil aromatic hydrocarbon (MOAH) content in olive pomace oils sampled at different points of the chemical refining process. Results (in mg/kg and with two significant figures) are given for the C-fractions and for the sum of them. Notation (i, m, f) corresponds to that of Table 1.

Sample	Refining	MOSH, mg/kg						MOAH, mg/kg				
#	Step	C10–C16	C16–C25	C25–C35	C35–C40	C40–C50	Total	C10–C16	C16–C25	C25–C35	C35–C50	Total
**2i**	raw	2.8	43	130	38	5.3	**219**	1.8	32	28	6.6	**68**
	neutralized	1.1	39	126	40	8.7	**216**	1.2	30	23	8.7	**63**
	winter./washed	1.2	40	119	51	7.4	**219**	1.3	29	18	13	**62**
	bleached	1.2	33	124	50	7.3	**215**	1.2	30	22	10	**64**
	refined	0	31	123	53	7.2	**213**	1.1	29	20	14	**63**
**2m**	raw	1.3	52	151	29	8.5	**241**	2.4	21	29	2.3	**54**
	neutralized	1.2	51	150	30	9.0	**241**	2.8	18	31	3.3	**55**
	winter./washed	1.2	50	150	31	8.9	**240**	2.5	20	31	2.7	**56**
	bleached	1.2	49	154	31	8.5	**243**	3.0	20	29	2.0	**55**
	refined	0	50	153	28	8.7	**240**	2.3	20	30	2.7	**55**
**3i**	raw	1.2	4.8	54	10	3.3	**73**	0	1.9	4.7	0	**6.6**
	neutralized	1.1	6.0	53	8.9	3.1	**72**	0	1.8	4.6	0	**6.4**
	winter./washed	1.2	6.5	52	9.1	3.7	**72**	0	2.1	4.8	0	**6.9**
	bleached	1.1	5.7	55	9.1	3.1	**74**	0	2.0	4.8	0	**6.8**
	refined	0	4.1	55	9.0	3.2	**72**	0	1.9	4.5	0	**6.5**
**7m**	raw	1.3	40	130	38	5.2	**215**	1.8	34	30	7.1	**73**
	neutralized	1.3	44	128	38	9.5	**221**	1.9	34	27	6.5	**69**
	winter./washed	1.4	42	132	38	8.7	**223**	1.8	37	23	6.7	**69**
	bleached	1.4	42	129	39	9.3	**221**	1.6	40	20	4.4	**66**
	refined	0	39	130	42	8.2	**220**	1.7	40	20	4.1	**66**
**9f**	raw	1.6	13	151	47	7.0	**220**	0	27	29	7.8	**63**
	winter./washed	1.5	17	151	55	8.5	**233**	0	24	27	7.2	**58**
	bleached	1.5	17	144	54	9.7	**226**	0	25	27	9.5	**61**
	refined	0	16	144	50	8.9	**219**	0	30	21	9.7	**61**
**10**	raw	1.5	18	76	16	6.5	**118**	0	10	14	4.1	**28**
	neutralized	1.5	19	74	15	7.9	**118**	0	10	16	3.3	**30**
	winter./washed	1.5	20	76	14	11	**123**	0	10	17	4.1	**31**
	bleached	1.4	20	78	14	10	**123**	0	11	16	3.4	**31**
	refined	0	20	69	15	12	**117**	0	11	18	3.2	**32**
**11**	raw	1.2	48	295	122	90	**556**	3.1	37	75	14	**129**
	neutralized	1.3	54	305	104	78	**542**	2.8	40	68	15	**125**
	winter./washed	1.3	53	306	104	78	**541**	3.1	39	73	17	**132**
	bleached	1.3	45	308	106	93	**553**	3.0	40	70	18	**131**
	refined	0	28	315	108	96	**547**	2.8	39	68	20	**131**

**Table 3 foods-12-00434-t003:** Mineral oil saturated hydrocarbon (MOSH) and mineral oil aromatic hydrocarbon (MOAH) content in commercial olive pomace oils. Results, in mg/kg and with two significant figures, are given for the C-fractions and for the sum of them.

Sample	MOSH, mg/kg					MOAH, mg/kg			
#	C20–C25	C25–C35	C35–C40	C40–C50	Total	C16–C25	C25–C35	C35–C50	Total
**1**	1.9	44	17	8.7	**71**	0	6.8	3.1	**9.9**
**2**	4.3	60	20	10	**94**	1.1	8.0	2.4	**12**
**3**	0	47	26	17	**91**	0	6.8	2.5	**9.3**
**4**	2.7	43	15	7.9	**69**	0	4.9	0	**4.9**
**5**	0	48	24	14	**86**	0	8.0	5.0	**13**
**6**	0	23	17	12	**52**	0	3.8	2.5	**6.4**
**7**	0	22	16	10	**48**	0	4.1	2.5	**6.6**
**8**	3.2	63	25	14	**105**	0	9.2	0	**9.2**
**9**	3.8	56	16	7.2	**83**	0	6.6	0	**6.6**
**10**	2.2	44	15	7.7	**69**	0	5.5	0	**5.5**
**11**	0	15	14	9.7	**39**	0	2.7	1.4	**4.1**
**12**	8.6	129	41	7.6	**186**	2.0	20	0	**22**
**13**	4.4	55	17	8.2	**85**	0	6.5	0	**6.5**
**14**	0	42	23	10	**76**	0	3.4	0	**3.4**
**15**	3.5	75	33	19	**131**	1.0	12	7.9	**20**
**16**	0	19	11	3.1	**33**	0	2.2	0	**2.2**
**17**	2.2	35	9.4	1.8	**49**	0	4.8	0	**4.8**
**18**	3.3	48	16	6.9	**74**	0	6.4	0	**6.4**
**19**	3.5	73	32	17	**126**	0	13	9.5	**23**
**20**	4.9	79	34	21	**139**	1.0	12	8.7	**22**
**21**	0	15	14	7.8	**37**	0	3.4	4.0	**7.4**
**22**	3.1	50	15	5.4	**73**	0	6.7	0	**6.7**
**23**	5.0	60	19	10	**94**	1.0	7.8	1.4	**10**
**24**	3.6	68	27	17	**116**	1.1	12	9.0	**22**
**25**	0	23	16	8.9	**49**	0	2.2	0	**2.2**
**26**	0	36	19	10	**65**	0	5.1	1.5	**6.6**
**27**	3.2	47	17	6.2	**73**	0	7.8	2.4	**10**
**28**	0	38	26	16	**79**	0	6.2	2.0	**8.3**
**29**	0	46	31	0	**77**	0	5.0	0	**5.0**
**30**	6.0	101	39	27	**173**	1.5	16	7.8	**25**
**31**	3.3	94	49	30	**176**	1.3	16	9.8	**28**
**32**	0	34	31	9.9	**74**	0	5.9	4.7	**11**
**33**	0	59	33	0	**93**	0	3.5	0	**3.5**
**34**	12	139	48	0	**199**	0	16	0	**16**
**35**	10	127	38	15	**190**	2.2	18	0	**20**
**36**	6.4	95	25	0	**126**	2.0	6.6	0	**8.7**
**37**	4.0	57	23	24	**108**	4.1	15	12	**31**
**38**	1.1	105	57	40	**203**	0	15	0	**15**
**39**	6.5	133	46	20	**205**	2.2	25	4.7	**32**
**40**	5.1	98	26	0	**129**	1.7	19	0	**21**
**41**	4.1	93	33	6.5	**136**	2.0	17	3.6	**23**
**42**	6.1	84	37	35	**161**	5.7	27	22	**55**
**43**	0	35	35	23	**94**	0	8.3	8.8	**17**
**44**	5.9	79	23	2.4	**110**	1.5	13	0	**15**
**45**	8.3	110	28	0	**146**	0	21	0	**21**
**46**	4.6	89	26	0	**120**	0	20	0	**20**
**47**	9.0	120	30	0	**159**	0	18	0	**18**
**48**	5.9	89	27	0	**122**	0	16	0	**16**
**49**	3.8	87	23	0	**114**	0	16	0	**16**
**50**	2.8	70	28	0	**101**	0	6.7	0	**6.7**
**51**	6.0	85	28	0	**119**	0	12	0	**12**

## Data Availability

The data presented in this study are available on request from the corresponding author. The data are not publicly available due to privacy restrictions.

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
