# Peer review of "The Mineral Oil Hydrocarbon Paradox in Olive Pomace Oils"

_foods, 2023, doi:10.3390/foods12030434_

Round 1
Reviewer 1 Report
The authors provide valuable data on MOH in olive pomace oils. This is of interest, since the levels are far above those presently considered as tolerable. Much has been done in the past to identify the source of this contamination (e.g. by Moret, who would deserve better citing in the Introduction), but so far this was unsuccessful – and also the new data now provided rather excludes some sources than identifying it. It is again pointing into the direction of an environmental contamination, but the levels are high for this. The authors could have added some calculations on the reconcentration by the residual oily in the pomace, but unfortunately they did not comment on this.
The work was performed with advanced technology. The manuscript is easy to read, but much of the result section consists of (correct) description of general knowledge that classically should be part of the Introduction. The language would profit from some improvement.
It would be important to specify how the olive oil was obtained and what its quality was, i.e. how the pomace was obtained. MOH contents in virgin oils tend to be higher than in extra virgin oils, which means that already a higher proportion of the MOH have been extracted. Is there such a correlation? Also a better description of how the pomace oils were obtained would be needed to meet requirements of science.
The authors seem to try to divert concerns from the high MOH level in olive pomace oils by saying that the MOSH are anyway not of concern and the MOAH do not include species with 3 and more aromatic rings. For this conclusion more quantitative data would be needed on the detection limit of the >2 ring MOAH (when MOAH concentrations are up to several 10 mg/kg). For MOSH, there is agreement that granulomas are probably not of concern (even though very frequent occurrence of a kind of granulomas was observed in the last century), but I doubt that the tox data available are sufficient to state that there is no concern for high olive pomace oil consumers also considering potential other end points. These conclusions need to be worded more carefully.
The title draws interest, but there is no explanation in the text why there is a paradox. Either “paradox” is explained in the text or the title needs modification.
There is a lack of reference to where the methods were from.
Some details by line number
9: toxicological assessment is not related to occurrence and limits are primarily determined by toxicology.
65: there are no fruits having/consisting of 90 % n-alkanes. Improve wording.
67: please add a reference on the distribution of the MOH in the olives, particularly for the MOH in the stones.
77: MOSH have shown to trigger granuloma formation in humans from the 1950es. To consider the formation of granulomas as the only possible consequence of exposure to MOSH is too short. In such a manuscript, a full toxicological evaluation is not needed, but it should not simplify.
135: please add a reference to this method
181: what is “criogen-free”?
186: a reference for the “reverse system” would be helpful.
249: reference to the work of Moret et al. is missing.
254: please shown the separation of the paraffins and the naphthenes or provide a reference.
267: the reference of Nestola & Schmidt and Biedermann et al (2020) are missing.
Table 1: For me, “chemical” implies a chemical reaction. Solvent extraction is not “chemical”; perhaps there is a better short term.
Figure 2: it should be explained that there was strong wrap around by overloading the second dimension column. Was the sample treated with alox?
356: MOSH-MOAH is MOH? Please clarify.
358: mg/kg hexane or related to oil?
360: did the solvent contain the MOH up to high masses? It is recovered by distillation.
380: comparting? comparing?
401: centrifugation to remove the water or to obtain the oil?
490: also from Bauwens et al.?
Figure 3A: where is this mixture from? B: why extracted ions rather than TIC as in A? what do the ions stand for?
544 (and elsewhere): by “endogenous” I tend to understand something like “natural” or formed by the plant. However, this is not the case. It is rather environmental – whatever is the path of the input.
555: among them there are carcinogenic ones – many are not carcinogenic
586: However, as shown by several papers, physical refining does remove a substantial part of the MOH, depending on the technology (unfortunately the distillate is usually used as additive to feeds).
602: the levels are only constantly high on olive pomace and grape seed oils. For the other oils mentioned, this was an incidental occurrence. This must be corrected.
Reviewer 2 Report
Contamination by MOSH and MOAH mineral oils in foods, including vegetable oils, is an interesting topic of considerable scientific and applicative interest. The proposed work is interesting and provides new information on the subject. Specifically, the following points can be listed.
1. This research is to define the level of contamination of mineral oils, such as MOSH and MOAH, in pomace oil crude and refined and at the same time, report some information on the concentration of MOSH and MOAH in commercial pomace oils.
2. I consider the topic is original and interest. The argument is try to give a scientific contribute to the European discussion about the limits, imposed by EFSA, for the mineral oils in olive oil categories
3. The research contains Information about the occurrence of mineral oils MOSH and MOAH in pomace olive oil and the impact of the refining process in the mineral oil concentration in pomace oils which add to the subject area compared with other published material.
4. The analytical methods are correct and extensively used to evaluated mineral oils and to define some preliminary information about chemical structure of these contaminants.
5. The conclusions are adequate, consistent with the evidence and arguments presented and do they address the main question posed.
6. The references are appropriate.
Author Response
Please see the attachement
